# Midwives' and obstetricians' perspectives about pregnancy related weight management in Ethiopia: A qualitative study

Fekede Asefa[1,2,3]*, Allison Cummins[2], Yadeta Dessie[1], Maralyn Foureu[2,4], Andrew Hayen[3]

1 School of Public Health, College of Health and Medical Sciences, Haramaya University, Harar, Ethiopia,
2 Centre for Midwifery, Child and Family Health, Faculty of Health, University of Technology Sydney, Sydney, NSW, Australia, 3 School of Public Health, Faculty of Health, University of Technology Sydney, Sydney, NSW, Australia, 4 Hunter New England Health, Nursing and Midwifery Research Centre, University of Newcastle, Newcastle, NSW, Australia

* sinboona@gmail.com

**Data Availability Statement:** Data can not be shared publicly because the interviews contain multiple personal identifier which the authors are not authorised to share. The data are available from

## Abstract

### Background

Midwives and obstetricians are key maternity care providers; they are the most trusted source of information regarding nutrition and gestational weight gain. However, their views, practices and perceived barriers to managing pregnancy related weight gain have not been studied in Ethiopia. The aim of this study was to explore midwives' and obstetricians' observations and perspectives about gestational weight gain and postpartum weight management in Ethiopia.

### Methods

We conducted face-to-face interviews with 11 midwives and 10 obstetricians, from January 2019 to March 2019. All interview data were transcribed verbatim. We analysed the data using thematic analysis with an inductive approach.

### Results

We identified three themes and associated subthemes. Midwives and obstetricians had limited knowledge of the optimal gestational weight gain. Almost all participants were unaware of the presence of the Institute of Medicine recommendations for optimal weight gain in pregnancy. According to the study participants, women in Ethiopia do not want to gain weight during pregnancy, but do want to gain weight after the birth. Counselling about gestational weight gain and postpartum weight management was not routinely provided for pregnant women. This is mostly because gestational weight gain counselling was not considered to be a priority by maternity care providers in Ethiopia.

Haramaya University via Research ethics committee (contact via Tel 0254662011; fax.0256668081; P.o.box 235)

**Funding:** An Australian Government Research Training Program provided a full scholarship for the corresponding author. This study was partially funded by the Haramaya University as a part of staff grant [grant number: HURG_2018_02_01_30]. The Centre for Midwifery, Child and Family Health, University of Technology Sydney provided financial support for data transcription purpose. There was no additional external funding received for this study.

**Competing interests:** The authors have declared that no competing interests exist.

**Abbreviations:** ANC, Antenatal care; BMI, Body Mass Index; GWG, Gestational Weight Gain; IOM, Institute of Medicine.

## Conclusions

The limited knowledge of and low attention to pregnancy related weight management by midwives and obstetricians in this setting needs appropriate intervention. Adapting a guideline for pregnancy weight management and integrating it into antenatal care is essential.

## 1. Introduction

Pregnant women are expected to gain weight due to physiological changes during pregnancy [1]. There are a number of gestational weight gain (GWG) guidelines [2, 3]. However, most of them were adapted from and similar to the 2009 United State Institutes of Medicine (IOM) recommendations guideline [2]. None of these guidelines were developed for Ethiopia or contexts similar to Ethiopia. The IOM recommends weight gain of 5 to 9 kg for obese women; 7 to 11 kg for overweight women; 11.5 to 16 kg for normal weight women, and 12.5 to 18 kg for underweight women [1].

Gaining gestational weight outside of the IOM recommendations can pose health risks for the baby [4] and the mother [5, 6]. Pregnant women who gain inadequate weight are at high risk of bearing a baby with low birth weight (LBW) and pre-term birth [7–10]. On the other hand, excessive GWG increases the risks of caesarean delivery [11], hypertension during pregnancy [5, 6, 10], postpartum weight retention [12], and development of long-term obesity [13]. Women are at a high risk of transitioning from normal weight to overweight or obesity following the pregnancy or during the postpartum period [13, 14]. Counselling pregnant women on gestational weight gain and dietary management increases the likelihood of gaining appropriate gestational weight [15, 16], and therefore helps to influence women's future health [13, 17].

Pregnancy related weight management is influenced by several factors [18, 19]. These factors include women's knowledge [18, 20], attitudes [19], and concerns [21] towards weight management during pregnancy; care providers' knowledge and confidence on GWG counselling [14, 22], perceived sensitivity of GWG by care providers [23–25], scepticism about the impact of counselling on women's behaviour [14], level of priority placed on GWG issues [14, 25, 26], shortage of time to discuss weight and nutrition during pregnancy care [27], and lack of familiarity with GWG guidelines [14, 26].

The 2016 Ethiopian Demographic and Health Survey reported that 13.4% of women of childbearing age were underweight; 57.2% were normal weight; 21.7 were overweight and 7.7% were obese; and almost half (49%) of these women were illiterate [28]. There are wide-ranging misconceptions and poor knowledge about nutrition during pregnancy [29]; and more than 70% (67–69% gain inadequate weight; 3–4% gain excess weight) of pregnant women in major cities of Ethiopia like Harar and Addis Ababa gain inappropriate gestational weight [30, 31].

Midwives and obstetricians are well positioned to advise pregnant women about how much weight to gain and how to manage nutrition and GWG appropriately during antenatal care visits [17, 32]. However, despite the 2016 World Health Organization (WHO) antenatal care (ANC) model recommending a minimum of eight ANC contacts [33], in Ethiopia the focused antenatal care model is still practised, resulting in around 90% of women in settings such as Addis Ababa receiving four ANC contacts (first contact before 16 weeks of pregnancy; second between 24 and 28 weeks of pregnancy; third at 32 weeks of pregnancy; and the fourth visit at 36 weeks of pregnancy) [34]. In the public health facilities of Addis Ababa, women with uncomplicated pregnancies receive care from midwives at the health centre level while women

with complicated pregnancies receive care from obstetricians at the hospital level. As key maternity care providers [35, 36], midwives and obstetricians are the most trusted source of information regarding nutrition and GWG [37].

Few studies have been conducted in Ethiopia focusing on GWG [4, 30, 31, 38]. No studies have addressed the issue of midwives' and obstetricians' views and practices regarding GWG and postpartum weight management. Given the influence of midwives [39, 40] and obstetricians [39, 41] on perinatal women, understanding their views, perceived barriers to managing GWG, and GWG management practices is important [42]. The aim of this study was to explore obstetricians' and midwives' views and practices related to GWG and postpartum weight management in this setting.

## 2. Methods

We used a qualitative descriptive study design. A qualitative descriptive approach provides a comprehensive understanding of given circumstance, a rich description of the participants' experiences or actions from the participants' perspectives [43]. Given midwives' and obstetricians' perspectives have not been explored in the Ethiopian context, a qualitative descriptive approach is an appropriate form of inquiry to describe how the midwives and obstetricians feel and manage GWG and postpartum weight retention. Therefore, we did not use a conceptual model or theory as a qualitative descriptive study aims to explore the who, what and where of midwives' and obstetricians' perspectives about pregnancy related weight management in Ethiopia. A qualitative descriptive approach is foundational to qualitative research and is a valuable methodological approach in and of itself without the need for a theoretical framework [44]. Moreover, qualitative descriptive findings can inform new interventions within the sociocultural context of the participants [45]. Ethics approval for the study was obtained from the Haramaya University Institutional Health Research Ethics Review Committee [IHRERC/200/2018], Addis Ababa Health Bureau Institutional Review Board [A/A/HB/2576/227] and the University of Technology Sydney, Human Research Ethics Committee [UTS HREC18-2610].

### 2.1 Study settings and participants

The study was conducted in Addis Ababa, the capital city of Ethiopia. Midwives and obstetricians, who provided maternity care services at the time of data collection in different health centres (seven health centres) and hospitals (four tertiary public hospitals) in the city, participated in the study.

**2.1.1 Recruitment.**   Midwives were purposively recruited from the seven health centres and one of the hospitals, while obstetricians were recruited from the four hospitals. Midwives and obstetricians who did not provide antenatal care services at the time of data collection were not included in the study (i.e., some of the midwives provide only labour and birth or postpartum care services. Similarly, some obstetricians were only engaged in teaching and surgery). Eligible participants were invited to participate in the study by the primary author. They were informed about the objectives of the study and the data collection procedure. Eleven midwives and 10 obstetricians agreed to participate in the study.

### 2.2 Data collection

Following written informed consent, we conducted 21 in-depth interviews, 11 with midwives and 10 with obstetricians, from January 2019 to March 2019. The primary author conducted the interviews in the local language, Amharic. The primary author is not a midwife or obstetrician, he is a public health professional and postgraduate student and did not have any influence over the participants. The data collection was performed through face-to-face interviews.

The interviews were held in a private room within the health facility (hospital or health centre), at a convenient time to the participants. We used open ended questions to elicit their views, perceptions and counselling experiences in relation to GWG and postpartum weight management. We developed the interview guide after reviewing the literature [14, 25, 46, 47]. The interview guide questions (both English and Amharic versions) are provided in the additional file (S1 Table). To test the interview questions, we conducted preliminary interviews with three midwives. Data collection continued until data saturation was reached.

Due to the qualitative nature of the study, we did not collect detailed demographic data about the participants other than to note professional affiliation, gender, employment location and years of work experience in maternity care (Table 1).

## 2.3 Data analysis

All audio recorded interviews were transcribed verbatim into English. The accuracy of transcripts was checked by comparing the text with each recorded interview, by the author (FA), who is a fluent Amharic and an English language speaker. We used Nvivo software version 11 (QSR International) [48] to manage written transcripts and to facilitate the coding process, categorising similar codes, and storage of the data. Due to practicality issues (i.e. data analysis was conducted in Australia), the data collected from the participants was not verified by the participants to determine whether the analysis of the data was consistent with the participants understanding of the comments made by them. The data was compared and contrasted by all authors to ensure rigour.

We analysed the data using thematic analysis [49] to assess repeated views, perspectives, and practices across all data. Data was collected until saturation was reached, that is hearing the same themes over and over. Different themes and sub-themes were developed from the data as described in Table 2 where an example of the coding is provided. During the analysis, we followed the six phases of thematic analysis according to Braun and Clarke's (2006) recommendations [50]. First, we began the analysis by reading and rereading to become familiar with the data and noted the main ideas from the data. Second, we examined transcripts line by line to identify dominant ideas and to draft codes. Third, we categorised similar codes into similar categories to search for possible themes and sub-themes. Fourth, we checked for the identified themes and sub-themes in relation to the coded extracts and the full data set. Fifth, we defined and named the themes and sub-themes while writing the overall findings that the analysis revealed. Finally, we developed the final report by selecting illustrative quotes. While using the quotes, participants were de-identified to maintain their anonymity and the quotes were presented in terms of participant numbers such as Obst.1 (obstetrician 1), or Mid.1 (midwife 1).

**Table 1. Summary of participant characteristics, Addis Ababa, Ethiopia, 2019.**

| Summary of participants | Midwives | Obstetricians | Total |
|---|---|---|---|
| Gender | | | |
| Female | 11 | 1 | 12 |
| Male | 0 | 9 | 9 |
| Years of experience | | | |
| 1 to 5 years | 2 | 7 | 9 |
| 5 to 10 years | 9 | 3 | 12 |
| 10 to 15 years | 1 | 0 | 1 |
| Place of work | | | |
| Health centre | 10 | 0 | 10 |
| Tertiary referral hospital | 1 | 10 | 11 |

**Table 2. Examples of themes, initial codes, and quotations, Addis Ababa, Ethiopia, 2019.**

| Main themes | Initial codes | Examples of quotations |
|---|---|---|
| Knowledge of optimal gestational weight gain | Expected weight gain; average 12.5 kg; availability of guideline; need a guideline | *A mother is expected to gain 10 to 15 kg and averagely 12.5 kg. . .* (Obst.3) |
| | | *The expected weight gain ranges from 0 to 12 kg* (Mid.1) |
| | | *. . .a mother should gain 10 to 12 kg during the whole pregnancy* (Mid.8) |
| | | *I do not know exactly. I have not seen it. I did not see the recent literature on weight gain. I know only about the need for additional weight gain, but I am not sure about IOM guideline.* (Obst.6) |
| | | *There is no guideline that focuses on GWG in our setup.* (Obst.9) |
| | | *We need a guideline, which focuses on our traditional foods.* (Obst.10) |
| Gestational weight gain counselling experience | No counselling; selective counselling; low attention; focusing on problems; low priority; high workload; focusing on problems | *We give attention only if she is underweight. We do not stress on it even in school. We are not giving attention to normal weight mothers. We do not consider [weight as important] as we do for headache or blood pressure.* (Obst.2) |
| | | *Most of the physicians neglect about the nutritional part and the subsequent weight gain.* (Obst.5) |
| | | *Usually, we do not talk to women about weight gain. Mostly we do not tell them but we advise them when their weight gets lower.* (Mid.4) |
| | | *Whether she is obese or not, we mention about nutrition. Other than this, we have never mentioned about weight gain intentionally.* (Mid.7) |
| | | *We consider about the weight gain only for extremely underweight mothers. You know we are busy with jobs and focus only on severely morbid [obese] mothers.* (Obst.2) |
| | | *We advise about nutrition in general, to eat what she has in her home. Otherwise, she could not afford. We simply advise her what she can get easily.* (Mid.6) |
| Ethiopian culture influences postpartum weight gain | Postpartum weight gain; weight related misconceptions; increased calorie intake; no physical exercise | *The main problem in our country is the dietary habit after birth. She eats porridge with butter. She will gain excessive weight.* (Obs.8) |
| | | *They think that gaining weight after birth as the natural process. We also do not advise them not to gain weight during the ANC visits.* (Obs.6) |
| | | *There is a time called 'Aras' after birth. Actually, they need additional energy at this time; they increase the food intake but there is no physical activity, so they gain excessive weight.* (Obs.4) |

## 3. Findings

We identified three themes and associated sub-themes: 1) knowledge of optimal gestational weight gain; 2) gestational weight gain counselling experience; 3) Ethiopian culture influences postpartum weight gain.

### 3.1. Knowledge of optimal gestational weight gain

All participants believed that women need to gain weight during pregnancy. However, their knowledge of the amount of weight gain in pregnancy varied amongst participants and varied when compared with the IOM recommendations. Obstetricians recommended pregnant women gain between 6 kg and 25 kg as described here:

*On average, a mother is expected to gain 12.5 kg throughout the pregnancy. . .[an] under-weight mother is expected to gain up to 16 kg. . . For obese mothers, they are expected to gain 6 to 7 kg.* (Obst.2)

*A mother is expected to gain 10 to 15 kg and on average 12.5 kg. . . if they have a BMI less than 18.5 [kg/m²], we expect a mother to gain up to 25 kg. If she had BMI greater than 30 or 40 [kg/m²], she may gain up to 7.5 kg.* (Obst.3)

We found similar variations in understanding amongst the midwives regarding the amount of gestational weight gain. Although obstetricians took account of women's BMI and explicitly explained the recommended amount of weight gain for underweight, normal weight, over-weight and obese women, most midwives discussed the recommended amount of GWG for normal weight women only.

*It is better if a mother gains 8 to 12 kilos [kg] in her pregnancy* (Mid.5)

*. . .they are expected to gain 13 to 20 kg* (Mid.3)

All study participants reported that there was no GWG guideline in Ethiopia, and the lack of a guideline contributed to their limited knowledge about GWG. When asked about the IOM guideline, only two obstetricians were aware of the presence of the IOM guideline. The sources of GWG information for study participants were pre-service training courses, books, and websites located using the Google search engine.

*There is no guideline that is prepared for us [in Ethiopia]. But we have gained this [GWG] information from our teachers, books and sometimes we read from Google. (*Mid.7)

All participants underscored the necessity of preparing a GWG guideline considering the local (Ethiopian) context. In addition to details about GWG, almost all obstetricians and some midwives stated the importance of including information such as the calorie and the nutrient contents of common local foods in the guideline.

*I suggest that the [GWG] standards should be prepared according to our context. . . (*Obst.2)

*The calorie content of foods should be [Ethiopian] specific. For example, we have to know the calorie content of single injera [an Ethiopian leaven flatbread with a slightly spongy texture] and shiro-wot [sauce made up of bean or pea powder mixed with different spices].* (Obst.3)

In addition, midwives expressed the need for training about GWG.

*There has to be training. If there is no training, it [GWG counselling] will not be my responsi-bility. . .If a health professional does not know [GWG information], who is going to be respon-sible if the mother gains or loses weight* (Mid. 3)

## 3.2. Gestational weight gain counselling experience

When asked about GWG counselling practices, many of the participants responded that they do not raise the issue of weight at all during antenatal care services, or that they do not provide women with specific information such as the appropriateness of the women's weight and the amount needing to be gained. Some participants believed that informing pregnant women about how much weight she has to gain or has gained was unnecessary. As discussed by this obstetrician,

*I don't think it is necessary to tell her and mention the amount of weight she has to gain. . .You cannot provide advice about the weight gained or she has to gain . . .They [pregnant women] also do not have much interest about weight gain.* (Obst.2)

In some circumstances the care providers did provide counselling particularly when a preg-nant woman did not gain weight or lost weight; presented with abnormal weight (underweight

or obese); or presented with problems such as hypertension or gestational diabetes mellitus. Some obstetricians outlined when they regarded it was important to provide counselling about weight gain;

*We consider discussing weight gain only for extremely underweight mothers. . . [We] focus only on severely morbid[ly obese] mothers. (Obst.2)*

*Sometimes we consider [GWG counselling]. . .if there is special condition like [when the woman is] underweight or very obese . . . (Obst.3)*

Midwives also explained that they only counselled a woman about weight gain when there was a comorbidity as described here:

*We mainly focus on medical cases. For example, if she is [a] DM [diabetes mellitus] or hypertensive patient . . . (Mid. 9)*

Almost none of the study participants provided counselling about gestational weight gain for women who had a normal weight. Described by the following midwives;

*We do not even talk about weight as long as her weight keeps within the normal range. We say 'your condition is good' and pass. (Mid. 6)*

*Most of the time, we do not advise them to measure their own weight and check it [regularly]. . . I do not think that they could do it because. . .Most of the time mothers work hard to overcome their life [challenges] and forget about the baby. (Mid. 4)*

Participants explained that counselling about gestational weight gain was a low priority and this led to four further sub-themes (gestational weight gain counselling was a low priority; not having enough time; confidence in providing GWG counselling; and protecting women from embarrassment).

**3.2.1. Gestational weight gain counselling was a low priority.** Most study participants reported that the main reason for not counselling about GWG was that GWG was not seen as a priority. As noted here;

*We focus on the vital sign measurements rather than weight gain. Therefore, we do not place much emphasis about maternal weight gain. It is not familiar [or common practice]. (Obst.1)*

Again this obstetrician stated it was a low priority unless the woman is under or overweight;

*It [GWG] is not something that attracts the attention of physicians. . .we give attention to the malnourished and extreme patient. (Obst.2)*

Advising women about GWG was not the midwives' usual practice either as this midwife stated;

*You may have other issues that you have to give priority to. You may give priority to BP [blood pressure] or any other danger signs. You do not waste time on this mother since it [weight gain] (Mid.2).*

Although they did not currently consider GWG counselling as a priority activity, most (nine midwives and eight obstetricians) believed that it should be a priority activity as this midwife explained;

*. . .nutritional counselling and weight gain issues must be a priority and [we] need to be focused. (Mid. 3)*

**3.2.2. Not having enough time.** Some participants reported that time constraints were a reason for not counselling about GWG. Both midwives and obstetricians expressed a lack of time to counsel about gestational weight gain;

*It is lack of time. A single physician examines lots of clients in the ANC clinic every day. So, we focus only on pertinent issues. (Obst.2)*

And again here;

*We have many clients; we do not even have time to discuss this [weight gain] (Mid. 2)*

Some midwives suggested that counselling about postpartum weight was not part of their scope of practice.

*. . .we do not tell them this [postpartum weight loss]. . .first, we are not taking this on as our job. Secondly, I do not know whether the society [will] accept you if you tell her to lose weight after childbirth. (Mid.1)*

A lack of women enquiring about postpartum weight advice, and high staff workload were raised as reasons for not counselling about postpartum weight management as this midwife described;

*The reason why we do not counsel them [on postpartum weight] is first because of the work burden on the health professional. . .The patient also does not ask about it and [this requires you to] initiate talking about this. (Mid.7)*

**3.2.3. Confidence in providing gestational weight gain counselling.** Most midwives felt they were not confident to advise about nutrition and GWG. They reported that their nutritional counselling discussions were shallow and did not address specific nutritional advice. As these midwives described;

*I do not think that I have full confidence . . .We do not tell her [a pregnant woman] that much in detail about her nutrition. . . (Mid.2)*

And again here

*As to me, I do not think that I am competent enough. Rather I might highlight to counsel them about issues related to nutrition. But this is not enough for the mothers. (Mid.3)*

Although obstetricians felt that they were relatively confident to advise about nutrition, they commented on the importance of involving nutritionists in perinatal counselling.

*It will be more effective if a nutritionist handled the case, because they may have detailed knowledge of the issue. Otherwise, if the nutritionist is not available . . . You may talk about the percentage of food groups needed, but not the amount of calories needed specifically from the nutritional groups. (Obs.3)*

**3.2.4. Protecting women from embarrassment.** Some study participants reported that the low economic status of women who attend ANC at public health facilities was perceived as a barrier to counselling on specific nutrition requirements. They felt that pregnant women would be embarrassed if they (women) could not afford to purchase what was advised to meet their nutritional requirements. Therefore, they counselled women to eat what they could access at home.

*We advise about the importance of nutrition in general, to eat what she has in her home. Otherwise, she could not afford [it]. We simply advise her to eat what she can easily get.* (Mid.6)

*I feel sad because . . .you cannot say take this much kilocalorie of apple or egg [the amount of calories found in an apple or egg]. . . Even we feel ashamed when we talk about food. So, we prefer to say [eat] any food in your home to cover this.* (Mid.9)

The most common nutritional counselling provided to pregnant women was to ensure the woman was aware of the need for an increase in the frequency of meals.

*I advise them to have frequent meals, like three times a day and additional snacks. . .* (Obs.3)

## 3.3. Ethiopian culture influences postpartum weight gain

From the midwives and obstetricians' perspectives, culturally, Ethiopian women consider that gaining weight after birth is a normal phenomenon. If a woman does not gain weight or if she loses weight during the postpartum period, the Ethiopian culture considers it as a sign of poor postpartum care. These obstetrician and midwife explained;

*They [women] consider gaining weight after delivery is a normal condition. . .. (*Obst.10)

*If she does not gain weight, culturally it is taken as if she did not get good care in [the] early postpartum period. . .* (Mid.7)

All study participants observed that most women gain weight during the postpartum period. After giving birth, compared to during pregnancy, almost all participants perceived eating more food of high energy density as the most common cause of postpartum weight gain.

*They gain more weight than [during the] pregnancy period. . .in our country, the situation is different. They consume more food during the post-partum [period] and may gain 20 to 30 kg.* (Obst.3)

*Usually they use [or consume] gruel, porridge and butter; and these foods have protein. This is our culture . . .* (Mid.2)

Participants stated the societal perception behind increasing food intake during the postnatal period is to replace blood and energy lost during the birth; to facilitate the healing process; and to increase milk production.

*Culturally it is believed that a woman loses lots of energy and excessive blood [while giving birth], and the society tries to replace the energy. They give [women] a high calorie diet until she returns to the pre-pregnancy state. . .* (Obst.2)

*It is perceived that she [has] had an injury and there is a bunch of meals provided from the society [friends and neighbours], in addition to the food prepared in the home.* (Obst.3)

*The main problem is myths. The myth is [that] the amount of food consumed and milk production is related. . . (*Obst.6)

All participants stated that there is a cultural expectation that a postpartum woman stays inside her home for some time (up to three months of the postnatal period), which could be one of the reasons for lack of exercise and this contributes to postpartum weight gain as this midwife described;

*. . .according to our country's culture, mothers are advised to get rest [stay at home] for at least 3 months. (*Mid. 9)

*Most of them don't lose weight due to lack of exercise. Especially, those who deliver through CS [caesarean section] [due to] fear that the sutures will be detached, they fear to engage in exercise.* (Obst.9)

## 4. Discussion

The present study explored the views, practices and observations of midwives and obstetricians regarding GWG and postpartum weight management. Study participants had limited knowledge regarding the optimal amount of gestational weight gain. Almost all participants were unaware of the existence of the IOM GWG recommendations. The participants discussed the need for a GWG guideline that is appropriate for Ethiopia. They observed the presence of widespread misconceptions among women about pregnancy weight management; that women did not want to gain weight during pregnancy but want to gain weight after birth. Participants reported that they did not provide counselling for pregnant women about weight management. The most common reason for the lack counselling was lack of attention to GWG and postpartum weight loss issues.

All participants reported that pregnant women need to gain weight due to physiological changes throughout pregnancy. However, they observed that women were not interested in gaining weight during pregnancy. Other studies in the major cities of Ethiopia like Harar and Addis Ababa have reported that more than 67% of pregnant women in Ethiopia gain below the IOM recommendations [30, 31]. This may be due to women decreasing food consumption during pregnancy perceiving that overeating or eating foods with high energy density may cause a large for gestational age baby that will make the birth difficult [29, 51] or that the IOM guidelines are not appropriate for this setting.

The amount of GWG recommendations stated by the study participants varies with each other and with the IOM recommendations. This could partly be explained by the lack of appropriate GWG guidelines for Ethiopia so that obstetricians use a variety of foreign sources such as textbooks and websites. By contrast, almost all midwives described a target weight gain for normal weight women only, and there was uncertainty among midwives regarding the description of the expected amount of GWG for underweight, overweight or obese women. Our finding is consistent with other studies in which midwives lack knowledge of the appropriate amount of GWG, even in high-income country settings [14, 22, 52]. This indicates that there is a need for improving understanding regarding the recommended amount of GWG, the importance of GWG and providing midwives and obstetricians with guidance to manage GWG [25].

Almost all study participants were not aware of the existence of or the recommendations in the IOM guideline. They suggested the need for developing GWG guidelines for Ethiopia.

Obstetricians stated the need for information about the nutrition and energy density of local foods, and the amount of energy intake needed during pregnancy, to be included in such a guideline. In addition, midwives lacked the confidence to offer nutritional counselling. Another study set in Ethiopia [53] recommended short-term in-service training for midwives to help them carry out nutrition and GWG counselling tasks.

Almost all study participants revealed that they routinely measure the weight of pregnant women but do not counsel appropriately on GWG. They either never raise the issue of weight at all or do not provide women with specific information such as how much weight they gained or need to gain. Some participants felt that telling women how much weight to gain is unnecessary. Most participants did not encourage women with normal weight to monitor their own weight. However, some counselled women about gestational weight when a woman either did not gain weight or had a loss of weight; had abnormal weight; or presented with conditions such as hypertension or gestational diabetes mellitus. Studies also revealed that physicians and prenatal specialists are more likely to counsel women who are at higher risk of or with disease than low-risk women [26, 27]. Midwives and obstetricians need to be encouraged to provide counselling about gestational weight gain management consistently for all pregnant women. The main reasons for lack of counselling about GWG in our study was a lack of knowledge and guidance in advice to provide women. Our participants also stated counselling around appropriate weight gain was a low priority. This finding is consistent with several other studies [14, 25, 26]. This indicates that midwives and obstetricians need to be well informed about the impact of inappropriate GWG on the health of the mother and baby so that they can appropriately prioritise the issue. Other reasons for lack of counselling was the high workload midwives and obstetricians experienced, and pregnant women in this setting did not generally enquire about their weight. However, while most of our study participants believed that initiating discussions about GWG was primarily their responsibility, they typically waited for the woman to raise this issue. Continuing education for midwives and obstetricians to develop skills in counselling women about nutrition during pregnancy and the postpartum period could help to improve the quality of maternal care provided for pregnant women in Ethiopia and ultimately reduce the workload for clinicians. An alternative approach could be to increase the health literacy of women and families in the community about the issue of GWG and the need to ensure pregnant and postnatal women receive optimal nutrition. Both of these recommendations would require resources and intervention at the level of government.

Midwives described a lack of confidence in advising women about GWG and nutrition as a barrier to their practice. This is consistent with other studies in which maternity care providers reported a lack of confidence in communicating about nutrition and weight as a major barrier to the management of GWG [14, 25, 54]. One study has shown that new midwifery graduates in Ethiopia have limited competency in nutrition counselling [53], which may be linked to inadequate nutrition education provided in midwifery programmes [52]. In contrast, although obstetricians felt that they were relatively confident to give nutritional advice, they underscored the necessity of involving nutritionists in antenatal counselling and the need for a GWG guideline. Studies also suggested a multidisciplinary approach, such as involving a nutritionist, in counselling and guidance of appropriate GWG [26, 32]. The most common nutritional advice that study participants provided for pregnant women was to increase their frequency of meals and to eat foods that were affordable or easily accessible at home. Counselling on specific nutrients and nutrition recommendations appeared to be even more difficult when the midwife or obstetrician believed the woman could not afford to buy specific foods. Affordability of appropriate nutrition is an issue that needs to be addressed by the Ethiopian government through improving pregnant women's income so that they can access and afford recommended foods during pregnancy.

In our study, all participants perceived that the postpartum weight management of Ethiopian women was inappropriate (i.e. most of the women aimed to gain weight during the postpartum period). According to the participants, the main factors that led women to gain weight during this period were an increase in food consumption and lack of physical exercise. Postpartum ceremonies include the family, relatives and even neighbours who prepare foods with high energy density such as porridge and gruel with butter for puerperal women. An increased intake of food during the postnatal period is aimed at replacing blood and energy lost during the birth; to facilitate the healing process for any injury that happened during childbearing; and to increase milk production. The participants believed that women's energy needs and their intake during the postpartum period were not balanced.

Participants stated that postpartum women do not engage in exercise and typically stay inside for up to three months of the postnatal period in Ethiopia. Confinement to the home is a common practice in other cultures too [55, 56], but the length of time is usually shorter (up to 40 days). Although a reasonable amount of rest is important to facilitate the transition to mothering, a prolonged time of decreased activity may lead to postpartum weight retention [57]. Therefore, postpartum women should be encouraged to take gentle exercise as early as appropriate following the birth [58].

Participants perceived that weight gain was encouraged during the post-partum period by most of the Ethiopian people and that women considered gaining weight after birth is a normal phenomenon. Losing weight or not gaining weight during the postpartum period was perceived as lack of proper care after birth. None of the participants provided proper postpartum weight management counselling for women. The participants reported several reasons related to lack of postpartum weight counselling. Some of these reasons–including a lack of attention to weight (postpartum weight); having a high workload; and a belief that women do not ask about their weight were similar to the reasons for lack of GWG counselling. Other reasons were the perception that postpartum weight counselling was not a common practice; postpartum weight counselling was not considered as their role; and fear that discussion of postpartum weight loss would be unacceptable to the Ethiopian society. Failure to address women's weight management during pregnancy, birth and postpartum period is a missed opportunity to influence a woman's future health [13, 59, 60]. According to the American College of Obstetricians and Gynaecologists, ensuring proper communication of postpartum issues with women is a responsibility of maternity care providers including obstetricians [17]. Since postpartum care or counselling is an ongoing process [17, 60], counselling on postpartum weight management should be started during pregnancy. Midwives and obstetricians require some ongoing education in regard to counselling around postpartum weight retention and information for pregnant women should be made available to ensure they understand the risks and benefits of inactivity and excessive weight gain in the postpartum period.

## 4.1. Strengths and limitations

Using a qualitative approach, we were able to explore midwives and obstetricians' views and practices around GWG, and barriers to counselling on GWG and postpartum weight management. This study is the first of its kind in Ethiopia. The complexity of interviewing in one language and translating the transcripts into another for analysis may have resulted in some issues being lost in translation [61]. However, we made a strenuous effort to check the accuracy of the translations by comparing the text with each recorded interview and the transcript. Another potential limitation is that this study was conducted in the capital city of Ethiopia; the situation in other parts of the country may be different.

## 5. Conclusions

This study explored the perspectives of midwives and obstetricians in Ethiopia regarding GWG and postpartum weight retention. The study found the awareness and practices of participants in relation to counselling pregnant women about appropriate weight gain were inconsistent. According to the midwives' and obstetricians' observations, there are widespread misconceptions about pregnancy weight management among women. Midwives lacked confidence to counsel women about GWG and nutrition whereas obstetricians considered other health issues to be a higher priority. The Ethiopian Ministry of Health (with concerned stakeholders such as the Ethiopian Society of Obstetricians and Gynecologists, and the Ethiopian Midwives Association) need to consider designing an education package or short-term in-service training concerning GWG for both midwives and obstetricians; adapting (preparing) a GWG guideline and integrating sufficient information about weight management into antenatal care guidelines as essential.

## Supporting information

**S1 Table. The topic guide for interviews with obstetricians and midwives, Addis Ababa, Ethiopia, 2019.**
(DOCX)

## Author Contributions

**Conceptualization:** Fekede Asefa, Allison Cummins, Yadeta Dessie, Maralyn Foureu, Andrew Hayen.

**Data curation:** Fekede Asefa, Yadeta Dessie, Maralyn Foureu.

**Formal analysis:** Fekede Asefa, Allison Cummins, Yadeta Dessie, Maralyn Foureu, Andrew Hayen.

**Funding acquisition:** Fekede Asefa, Yadeta Dessie.

**Methodology:** Fekede Asefa, Allison Cummins, Yadeta Dessie, Maralyn Foureu, Andrew Hayen.

**Supervision:** Allison Cummins, Yadeta Dessie, Maralyn Foureu, Andrew Hayen.

**Writing – original draft:** Fekede Asefa.

**Writing – review & editing:** Fekede Asefa, Allison Cummins, Yadeta Dessie, Maralyn Foureu, Andrew Hayen.

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
