## [Decision Letter · Decision Letter 0]

22 Oct 2020

PONE-D-20-24673

Midwives’ and Obstetricians’ Perspectives about Pregnancy Related Weight Management in Ethiopia: A qualitative study

PLOS ONE

Dear Dr. Fekede Asefa,

Thank you for submitting your manuscript to PLOS ONE. After careful consideration, we feel that it has merit but does not fully meet PLOS ONE’s publication criteria as it currently stands. Therefore, we invite you to submit a revised version of the manuscript that addresses the points raised during the review process.

We look forward to receiving your revised manuscript.

Kind regards,

Sharon Mary Brownie

Academic Editor

PLOS ONE

Journal Requirements:

2. You have indicated in the submissions page that "all relevant data are within the manuscript and its Supporting Information files"; however, there  are no data in the supporting files. Please submit these data as Supporting files.

3. Please include additional information regarding the interview guide used in the study and ensure that you have provided sufficient details that others could replicate the analyses. For instance, if you developed a guide as part of this study and it is not under a copyright more restrictive than CC-BY, please include a copy, in both the original language and English, as Supporting Information.

4. Please clarify whether a conceptual model or theory was used for this research. If not, please provide a rationale for not doing so.

Reviewers' comments:

Reviewer's Responses to Questions

**Comments to the Author**

1. Is the manuscript technically sound, and do the data support the conclusions?

Reviewer #1: Yes

Reviewer #2: Yes

Reviewer #3: Yes

2. Has the statistical analysis been performed appropriately and rigorously? 

Reviewer #1: N/A

Reviewer #2: N/A

Reviewer #3: Yes

3. Have the authors made all data underlying the findings in their manuscript fully available?

Reviewer #1: No

Reviewer #2: Yes

Reviewer #3: Yes

4. Is the manuscript presented in an intelligible fashion and written in standard English?

Reviewer #1: Yes

Reviewer #2: Yes

Reviewer #3: No

5. Review Comments to the Author

Reviewer #1: Comments

1. Due to practicality issues (i.e. data analysis was conducted in Australia), the data collected from the participants was not verified by the participants to determine whether the analysis of the data was consistent with the participants understanding of the comments made by them.

2. The analysis does not imply an inductive approach to data analysis. It seems to be more deductive because you do not mention any pre-determined ideas regarding your themes and codes.

3. I am not so sure how you did the actual coding maybe availing the codebook would be of help. In Table 3: the way the codes are named does not seem right. For example, naming a code absence of guidelines in more or less a biased code. I think it is better to name it ‘availability of guidelines’ because participants may not necessarily talk about the absence of guidelines but have broader discussions on the availability of guidelines.

4. The subtitles of your results are biased and may not give a true picture of your analysis. For example, if you say “having limited knowledge of optimal GSW” I wonder if all the participants’ narratives indicated limited knowledge. The same applies to “lacking skills in counselling” I would suggest you name it “counselling experiences…” instead of lack of self-confidence to providing counselling you may call it “self-efficacy”

5. What are the health implications of GWG??? That is for both the mother and baby. Are there any public health implications pertaining to maternal and child health outcomes? This could be indicated in the background of the study.

Reviewer #2: key words: counseling in the context of the text is vague and should removed, post-partum weight retention should be changed post-partum weight counseling.

METHODS: What informed the authors about the choice of the facilities the recruited the participants from? It is not clear how they selected the participants from the various facilities and determined their number. Why were the authors so particular about the United states institute of medicine's recommendations on weight gain in pregnancy, has this been adopted in the study country of in countries with similar population characteristics in the region or maternal indices?

Reviewer #3: In this study entitled: Midwives’ and Obstetricians’ Perspectives about Pregnancy Related Weight Management in Ethiopia: A qualitative study

1The study, context is in keeping with other literature on the topic. Physiological changes take place during pregnancy leading to weight gain. Therefore women are at a high risk of being overweight to obesity during postpartum period

2 since there is misunderstanding and inadequate knowledge of nutrition during pregnancy women need to be counseled about (GWG) and nutritional care in order to maintain adequate gestational weight. Midwives and obstetricians take care of pregnant women depending on the complication and facility therefore is in position to advice on GWS

3 The authors therefore set out to explore obstetricians’ and midwives’ views and practices related to GWG and postpartum weight management in this Ethiopian setting

4. The population was appropriate for the study since midwives and obstetricians are key maternity care providers and they are the most trusted source of information regarding nutrition and gestational weight gain.

5 The qualitative descriptive study design with face to face interview method with open ended questions used was appropriate

7 The authors analyzed data by thematic analysis with inductive approach which is appropriate for qualitative studies

8 Presentations of data available is appropriate with tables supporting it

10According to the authors there was inconsistence in knowledge and practice of counseling in gestational weight gain. Midwives lacked confidence to counsel the women while the obstetricians had other priority health issues

Reviewer’s comments

Generally there are English grammars mistakes that the authors need to address using grammarly or seek other assistance

Please clearly re-write the title for table 1

What was the criterion for selecting the health facilities in the city. Are they the only maternal health care faculties available?

How was the sample size determined?

Although midwives and obstetricians are the key maternity care providers why didn’t the author consider other healthcare givers who may be involved in maternal health care?

6. PLOS authors have the option to publish the peer review history of their article (what does this mean?). If published, this will include your full peer review and any attached files.

Reviewer #1: No

Reviewer #2: No

Reviewer #3: **Yes: **ESTER LILIAN ACEN

---

## [Author Response · Author response to Decision Letter 0]

2 Dec 2020

First of all, we would like to thank the editor and the reviewers for the gracious support with their time and inputs. The insightful comments and suggestions have resulted in a much improved manuscript. Please kindly find below the point-by-point responses to instructions and suggestions from the editor and reviewers. Thank you all again for considering our manuscript.

We thank the editor for the instruction. We have ensured our manuscript followed PLOS ONE's style requirements.

2. You have indicated in the submissions page that "all relevant data are within the manuscript and its Supporting Information files"; however, there are no data in the supporting files. Please submit these data as supporting files.

We thank the editor for the comment. We have revised the summations page to indicate the data will provide data upon request to the Haramaya University via ethics committee for eligible researchers. 

3. Please include additional information regarding the interview guide used in the study and ensure that you have provided sufficient details that others could replicate the analyses. For instance, if you developed a guide as part of this study and it is not under a copyright more restrictive than CC-BY, please include a copy, in both the original language and English, as Supporting Information.

We thank the editor for the comment. We have submitted the interview guide as additional information. 

4. Please clarify whether a conceptual model or theory was used for this research. If not, please provide a rationale for not doing so.

We thank the editor for raising this point, which we agree will benefit from more clarification. We used a qualitative descriptive approach to help us gain insight into the perspectives and experiences of Midwives’ and Obstetricians regarding gestational weight gain and postpartum weight management. Therefore, we did not use a conceptual model or theory as a qualitative descriptive study aims to explore the who, what and where of Midwives’ and Obstetricians’ Perspectives about Pregnancy Related Weight Management in Ethiopia. A qualitative descriptive approach is foundational to qualitative research and is a valuable methodological approach in and of itself (Sandelwoski 2000) without the need for a theoretical framework. We have now included this information in the paper. 

Reference 

Sandelowski M. Focus on Research Methods Whatever Happened to Qualitative Description? Res Nurs Health.2000;23:334-40.

Reviewer's Responses to Questions

Reviewer #1:

1. Due to practicality issues (i.e. data analysis was conducted in Australia), the data collected from the participants was not verified by the participants to determine whether the analysis of the data was consistent with the participants understanding of the comments made by them.

Thank you for your comments, data was collected in Ethiopia and due to time constraints with travel back to Australia for data analysis we were unable to ask the participants to verify the analysis. We followed the Braun and Clarke (Braun and Clark 2006) six phase approach to thematic analysis and this involved all authors comparing and contrasting the findings to ensure rigour. 

Reference 

Braun V, Clarke V. Using thematic analysis in psychology. Qualitative Research in Psychology. 2006;3 (2):77-101.

2. The analysis does not imply an inductive approach to data analysis. It seems to be more deductive because you do not mention any pre-determined ideas regarding your themes and codes.

We thank Reviewer #1 for point out this issue. This is an error we have removed and replaced with data were collected until data saturation was reached, that is hearing the same themes over and over. We used the Braun and Clarke six phase approach for analysis (Braun and Clark 2006). First, we began the analysis by reading and rereading to become familiar with the data and noted the main ideas from the data. Second, we examined transcripts line by line to identify dominant ideas and to draft codes. Third, we categorised similar codes into similar categories to search for possible themes and sub-themes. Fourth, we checked for the identified themes and sub-themes in relation to the coded extracts and the full data set. Fifth, we defined and named the themes and sub-themes while writing the overall findings that the analysis revealed. Finally, we developed the final report by selecting illustrative quotes.

Reference 

Braun V, Clarke V. Using thematic analysis in psychology. Qualitative Research in Psychology. 2006; 3 (2):77-101.

3. I am not so sure how you did the actual coding maybe availing the codebook would be of help. In Table 3: the way the codes are named does not seem right. For example, naming a code absence of guidelines in more or less a biased code. I think it is better to name it ‘availability of guidelines’ because participants may not necessarily talk about the absence of guidelines but have broader discussions on the availability of guidelines.

We thank the reviewer for the important comment. We revised the codes in Table 3 according to your suggestion. We changed the code ‘absence of guidelines’ to ‘availability of guidelines’. 

4. The subtitles of your results are biased and may not give a true picture of your analysis. For example, if you say “having limited knowledge of optimal GSW” I wonder if all the participants’ narratives indicated limited knowledge. The same applies to “lacking skills in counselling” I would suggest you name it “counselling experiences…” instead of lack of self-confidence to providing counselling you may call it “self-efficacy”

We thank Reviewer #1 for the suggestions. To make theme one more inclusive, we changed “having limited knowledge of optimal GWG” to “knowledge of optimal gestational weight gain”. Similarly we have changed “lacking skills in counselling about gestational weight gain” to “gestational weight gain counselling experience”. We felt that ‘self-efficacy’ would be a bit out the context, therefore, we changed “lack of self-confidence to providing counselling” to “confidence in providing GWG counselling”.

5. What are the health implications of GWG??? That is for both the mother and baby. Are there any public health implications pertaining to maternal and child health outcomes? This could be indicated in the background of the study.

We thank Reviewer #1 for the comment. We have now included the health implications of GWG for both the mother and the baby. It is described in paragraph two. 

Reviewer #2:

1. Key words: counselling in the context of the text is vague and should removed, post-partum weight retention should be changed post-partum weight counselling.

We thank Reviewer #2 for the suggestions. 

1. We agree that the term counselling is vague. We have replaced “counselling” with more specific terms “gestational weight gain counselling”

2. We also agree that the term “post-partum weight retention” does not reflect the finding. We have followed your suggestion and changed the term “post-partum weight retention” to “post-partum weight counselling”

2. METHODS: What informed the authors about the choice of the facilities the recruited the participants from? It is not clear how they selected the participants from the various facilities and determined their number. 

We thank Reviewer #2 for the questions. We selected the health facilities purposely. This study was part of a mixed method study in which the quantitative aspect was intended to assess patterns and predictors of GWG among pregnant women. The midwives and pregnant women recruited to this qualitative part of the study were selected from the same health centres. Obstetricians were selected from tertiary hospitals at which most of the obstetricians in the city currently provide obstetric care and teaching services. 

The number of participants was determined based on data saturation, i.e. when new ideas, perspectives and explanations were no longer heard during the interview or the discussion, and variation in the data was levelling off, data saturation was evident (Saunders et al., 2018; Fusch and Ness, 2015).

Reference 

Saunders B, Sim J, Kingstone T, Baker S, Waterfield J, Bartlam B, Burroughs H, Jinks C. Saturation in qualitative research: exploring its conceptualization and operationalization. Qual Quant. 2018; 52(4): 1893–1907.

Fusch P and Ness L: Are We There Yet? Data Saturation in Qualitative Research. The Qualitative 2015; 20 (9): 1408-1416

Why were the authors so particular about the United states institute of medicine's recommendations on weight gain in pregnancy, has this been adopted in the study country of in countries with similar population characteristics in the region or maternal indices?

We thank Reviewer #2 for raising this critical point. There are a number of GWG guidelines. However, most of them were adapted from and similar to the 2009 United State Institutes of Medicine (IOM) recommendations (Alavi 2013; Scott 2014). None of these guidelines were developed for Ethiopia or contexts similar to Ethiopia. In the absence of any guideline for Ethiopia or other low-income settings, we decided to use the most widely accepted guideline, the IOM guideline, from which other GWG guidelines were adapted. We recommended development of a GWG guideline which could be suitable to the Ethiopian context based on our findings. This was described in the discussion and conclusion sections of the manuscript. 

References

Alavi N, Haley S, Chow K, McDonald SD. Comparison of national gestational weight gain guidelines and energy intake recommendations. Obesity Review. 2013; 14:68–85.

Scott C, Andersen CT, Valdez N, Mardones F, Nohr EA, Poston L, et al. No global consensus: a cross sectional survey of maternal weight policies. BMC Pregnancy Childbirth. 2014; 14(167):1–10.

Reviewer #3: In this study entitled: Midwives’ and Obstetricians’ Perspectives about Pregnancy Related Weight Management in Ethiopia: A qualitative study

1. The study, context is in keeping with other literature on the topic. Physiological changes take place during pregnancy leading to weight gain. Therefore women are at a high risk of being overweight to obesity during postpartum period. 

We thank Reviewer #3 for summarising the points. 

2. Since there is misunderstanding and inadequate knowledge of nutrition during pregnancy women need to be counselled about (GWG) and nutritional care in order to maintain adequate gestational weight. Midwives and obstetricians take care of pregnant women depending on the complication and facility therefore is in position to advice on GWS

We thank Reviewer #3 summarising the points. 

3. The authors therefore set out to explore obstetricians’ and midwives’ views and practices related to GWG and postpartum weight management in this Ethiopian setting

We thank Reviewer #3 summarising the points. 

4. The population was appropriate for the study since midwives and obstetricians are key maternity care providers and they are the most trusted source of information regarding nutrition and gestational weight gain.

We thank Reviewer #3 summarising the points.

5. The qualitative descriptive study design with face to face interview method with open ended questions used was appropriate

We thank Reviewer #3 for the comment. 

6. The authors analyzed data by thematic analysis with inductive approach which is appropriate for qualitative studies

We thank Reviewer #3 for the comment. 

7. Presentations of data available is appropriate with tables supporting it

We thank Reviewer #3 for the comment. 

8. According to the authors there was inconsistence in knowledge and practice of counselling in gestational weight gain. Midwives lacked confidence to counsel the women while the obstetricians had other priority health issues

We thank Reviewer #3 for the comment. 

Generally there are English grammars mistakes that the authors need to address using grammarly or seek other assistance. 

We thank Reviewer #3 for the comment. We have conducted another proof-read of the paper and tried to correct and avoid grammatical mistakes

Please clearly re-write the title for table 1

We thank Reviewer #3 for the suggestion. We have re-written the title for table 1

What was the criterion for selecting the health facilities in the city. Are they the only maternal health care faculties available?

We thank Reviewer #3 for the questions. We selected the health facilities purposely. This study was part of a mixed method study in which the quantitative aspect was intended to assess patterns and predictors of GWG among pregnant women. The midwives and pregnant women recruited to this qualitative part of the study were selected from the same health centres. Obstetricians were selected from tertiary hospitals at which most of the obstetricians in the city currently provide obstetric care and teaching services. 

How was the sample size determined?

We thank Reviewer #3 for the questions. The number of participants was determined based on data saturation, i.e. when new ideas, perspectives and explanations were no longer heard during the interview or the discussion, and variation in the data was levelling off, data saturation was evident (Saunders et al., 2018; Fusch and Ness, 2015).

Reference 

Saunders B, Sim J, Kingstone T, Baker S, Waterfield J, Bartlam B, Burroughs H, Jinks C. Saturation in qualitative research: exploring its conceptualization and operationalization. Qual Quant. 2018; 52(4): 1893–1907.

Fusch P and Ness L: Are We There Yet? Data Saturation in Qualitative Research. The Qualitative. 2015; 20 (9): 1408-1416

We thank the editor and all Reviewers again for the important inputs in the manuscript. Please do not hesitate to let us know if there are any further comments or suggestions.

---

## [Editor Report · Decision Letter 1]

7 Dec 2020

Midwives’ and Obstetricians’ Perspectives about Pregnancy Related Weight Management in Ethiopia: A qualitative study

PONE-D-20-24673R1

Dear Dr.Fekede Asefa,

We’re pleased to inform you that your manuscript has been judged scientifically suitable for publication and will be formally accepted for publication once it meets all outstanding technical requirements.

Kind regards,

Sharon Mary Brownie

Academic Editor

PLOS ONE

---

## [Editor Report · Acceptance letter]

9 Dec 2020

PONE-D-20-24673R1 

Midwives’ and Obstetricians’ Perspectives about Pregnancy Related Weight Management in Ethiopia: A qualitative study 

Dear Dr. Asefa:

I'm pleased to inform you that your manuscript has been deemed suitable for publication in PLOS ONE. Congratulations! Your manuscript is now with our production department. 

Kind regards, 

on behalf of

Professor Sharon Mary Brownie 

Academic Editor

PLOS ONE